# The Impact of the COVID-19 Pandemic on the Mental Health of Teachers and Its Possible Risk Factors: A Systematic Review

**DOI:** 10.3390/ijerph20031747

**Published:** 2023-01-18

**Authors:** Iago Sávyo Duarte Santiago, Emanuelle Pereira dos Santos, José Arinelson da Silva, Yuri de Sousa Cavalcante, Jucier Gonçalves Júnior, Angélica Rodrigues de Souza Costa, Estelita Lima Cândido

**Affiliations:** 1School of Medicine, Federal University of Cariri (UFCA), Barbalha 60430-160, Brazil; 2Division of Rheumatology, Faculdade de Medicina FMUSP, Universidade de São Paulo, São Paulo 05403-010, Brazil; 3Programa de Pós-Graduação em Desenvolvimento Regional Sustentável (PRODER), Federal University of Cariri (UFCA), Juazeiro do Norte 63048-080, Brazil

**Keywords:** COVID-19, mental disorders, mental health, school teacher, teaching

## Abstract

(1) Objective: The objective was to analyze the development of psychiatric pathologies/burnout syndrome and their possible risk factors in teachers in the context of the COVID-19 pandemic. (2) Methods: A qualitative systematic review was carried out, according to the PRISMA protocol, in the PubMed, Scopus, and Web of Science databases using a combination of the following descriptors [MeSH]: “mental health”, “mental disorders’’, “covid-19” and “school teachers’’. Articles selected were written in English, Portuguese and Spanish, published between November 2019 and December 2022. (3) Results: The most common psychiatric pathologies were generalized anxiety disorders and depression. Burnout syndrome was also quite prevalent. Of the 776 articles identified, 42 were selected after applying the eligibility criteria. Although there is variability among the analyzed studies, the risk factors most correlated with increased morbidity in teachers were: (i) being female; (ii) age below the fifth decade of life; (iii) pre-existence of chronic or psychiatric illnesses before the pandemic; (iv) difficulty in adapting to the distance education model; (v) family/work conflicts; (vi) negative symptoms caused by the pandemic. (4) Conclusions: Therefore, the COVID-19 impact on mental health appears to be more common in female teachers in their fifth decade of life and with pre-existing psychiatric comorbidities. However, prospective studies are needed to better map this situation.

## 1. Introduction

The literature has shown that there are higher levels of psychiatric illnesses such as depression, anxiety disorders, loss of quality of life, fatigue and negative feelings such as stress and frustration in teachers before the COVID-19 pandemic. Studies carried out in Brazil [1], Germany [2,3] and UK [4] corroborate this data.

During the COVID-19 pandemic, the fear of being infected, the growing number of cases and deaths, the sanitary measures necessary to fight the pandemic (e.g., social distancing and quarantine) [5] increased the deterioration of mental health by promoting the onset or worsening of psychiatric disorders [6,7] in the general population. Thus, in professions known to be exposed to high levels of stress, such as teachers [8,9,10], the pandemic may have the potential to worsen psychiatric pathologies, negative feelings or trigger them [11,12].

This can happen because COVID-19 created new challenges for teachers such as having to quickly adapt face-to-face teaching to a remote environment [11]. Changes in the way teachers work, without adequate training for the new teaching modality, have generated an unprecedented challenge for education professionals, such as the difficulty of using digital platforms, the lack of resources to teach remote classes, work overload, and the excessive use of screens [12]. In addition, the incorporation of different teaching strategies such as instant messaging via apps created the illusion that teachers were always available for work, making it difficult to reconcile personal and professional responsibilities [7].

Thus, in view of this challenging scenario, we ask: what is the impact of COVID-19 on the mental health of teachers and what possible risk factors for this population may be associated with worse mental health indices?

Our hypothesis is that factors that are known to be stressors for the general population, such as the impact of social isolation, in different ways may trigger mental illness and burnout syndrome at work in teachers [3,4,5] due to the previously mentioned particularities [7,11,13].

Thus, it is necessary to map the real impact of the COVID-19 pandemic on teachers, so that public policies can be formulated along with the competent agencies of each country, aiming at support plans for these professionals, as well as supporting future prospective studies that aim at long-term discrimination of the impact of COVID-19 on the mental health of teachers [14,15].

Therefore, the objective was to analyze the development of psychiatric disorders (generalized anxiety disorder (GAD), post-traumatic stress disorder (PTSD) and depression) and burnout syndrome (BS) and its possible risk factors in teachers of all levels of education in the context of the COVID-19 pandemic.

## 2. Material and Methods

### 2.1. Literature Review

A qualitative systematic review was conducted to investigate the occurrence of psychiatric disorders (generalized anxiety disorder (GAD), post-traumatic stress disorder-(PTSD) and depression) and burnout syndrome (BS) in teachers, from preschool to higher education, who have been working during the COVID-19 pandemic. For this, the PubMed, Scopus, and Web of Science databases were consulted in February 2022 and December 2022. The search period was between 1 November 2019 and 1 December 2022, using the descriptors “mental health” (MeSH -Medical Subject Headings), “mental disorders” (MeSH), “covid-19” (MeSH) and “school teachers” (MeSH), combined with the Boolean operators AND and OR.

### 2.2. Data Collection and Eligibility Criteria

The analysis of articles in the databases was conducted by two investigators to assess the inclusion of studies, and discrepancies were weighted by a third investigator. Initially, titles and abstracts were analyzed based on the following inclusion criteria: approach to the investigated topic, explicit methodology, adequate temporality, and texts in English, Portuguese or Spanish. Systematic reviews, letters, case reports, book chapters, and purely qualitative studies were excluded from the review.

From the selected studies, detailed information was extracted, such as the design, the number of subjects, the teacher’s education level, the country in which the study was carried out, the disorders evaluated, the scales used, and the main results (Table 1).

### 2.3. Assessment of Methodological Quality

The methodological quality of the studies selected in our review was analyzed using the Newcastle Ottawa Scale (NOS), adapted for cross-sectional studies [13]. The original instrument aims to assess the methodological characteristics of longitudinal studies, specifically cohorts and case-controls. In our study, we used the NOS adapted for cross-sectional studies, whose biggest change consists in changing the third principle from “Exposure” to “Outcome”. Thus, three main domains were evaluated: Selection, Comparability, and Outcome. In the first aspect, the representativeness of the cases, the size of the sample, the description of the rate of non-respondents, and the method for determining exposure are evaluated. In “Comparability”, it is observed whether there is control of confounding factors. Finally, in “Outcomes”, the form of access to the outcome and the statistical test used are evaluated. Respectively, each domain can receive a maximum of five, two, and three points, totaling ten points. In our evaluation, the topic that scored zero did not receive any points. For the longitudinal studies present in this review, we chose to use the same adapted scale given the similarity of the methodological design of the selected longitudinal studies with the cross-sectional ones.

As a way to improve the quality analysis of studies and based on the convenience of the present study’ authors, the Quality Assessment Tool for Observational Cohort and Cross-Sectional Studies was used. This is a tool linked to the study Quality Assessment Tool (https://www.nhlbi.nih.gov/health-topics/study-quality-assessment-tools, accessed 13 November 2022)—National Heart, Lung and Blood Institute (NHLBI). These tools classify studies as “Good”, “Fair” or “Poor”, based on the presence or absence of relevant methodological elements for each type of study (Table 1). According to this scale, if there was presence of the methodological characteristic, the work scored ‘yes’ (Y), if there was ‘no’ (N) and if it did not belong to the scope of the methodological type, it received ‘not applicable’ (NA).

### 2.4. Characteristics of the Studies Included

A total of 926 articles were identified, of which 776 were screened after removal of duplicates. Seven hundred and fourteen papers were excluded after analyzing the titles and abstracts, as well as due to the lack of relationship with the topic, unavailability, and diverse methodology (review, short communication, comment, opinion, perspective and theoretical studies). Of the 62 final articles, 13 articles were excluded because they had a qualitative analysis, and seven due to the unclear description of the methodology used. Thus, our final sample consisted of 42 articles (Figure 1).

Thirty-seven articles were cross-sectional, while five were longitudinal studies. The population investigated in the studies ranged from 55 to 88,611 teachers, resulting in 227,475 subjects. Most of the studies were carried out in Europe (*n* = 14). Teachers from countries on the Asian and American continents were investigated in eleven and twelve articles, respectively, with the US (*n* = 7) and China (*n* = 7) having the largest number of articles from each continent. Only one study [18] evaluated the context of psychological illness of teachers during the pandemic on six continents. One carried out an investigation in two countries (Spain and Ecuador).

The tools used by the authors ranged from author questionnaires to established instruments. Among the author questionnaires, the Likert Scale was the most used, with variation concerning the total points per item (from four to six points). The most used validation instruments were: Maslach Burnout Inventory, GHQ-12, DASS-21, PHQ-9, and GAD-7.

Fourteen studies evaluated teachers at various educational levels (from preschool to higher education). Fourteen studies investigated elementary, middle school and high school teachers, five evaluated high school teachers, three evaluated elementary and middle school teachers and high school teachers, and six evaluated university professors. Table 1 and Table 2 shows the characteristics of the articles included in the systematic review.

### 2.5. Assessment of Methodological Quality

Most of the articles scored seven points or more on the Newcastle Ottawa Scale. Four studies had the highest score, while seven received nine points and nine articles, eight points. Eight studies scored six points and thirteen studies scoring seven points. Only one article obtained five points, the lowest score in the evaluation (Figure 2).

Regarding the analysis of Quality Assessment Tool for Observational Cohort and Cross-Sectional Studies, most studies were classified as “Good” (*n* = 38) and four studies were classified as “Fair”. As for the criteria analyzed, most of the articles did not present data on the participation rate of elected professors or did not demonstrate a justification for the size of the chosen population. All had adequate measures of exposure and outcome.

### 2.6. Ethical Issue

Considering this is a systematic literature review, Resolution 510/16 of the Brazilian National Health Council (CNS, acronym in Portuguese) ensures the dismissal of the submission to an Ethics Committee on Research (Human Beings). This review is registered on the PROSPERO Platform (https://www.crd.york.ac.uk/prospero/, accessed on 15 January 2023) under the number CRD42022381294.

## 3. Results

### 3.1. Generalized Anxiety Disorder, Teaching and the COVID-19 Pandemic

The generalized anxiety disorder (GAD) studies included in this review [10,13,16,18,20,21,24,25,26,27,29,31,34,36,37,38,40,41,42,44,45,46,49,50,51] documented the presence of GAD symptoms among teachers. Those who investigated the prevalence of symptoms showed a range from 38.4% to 73% between the professionals involved. The values varied according to the population studied and the instrument used. Akour et al. [16] and Lizana and Lera [34] presented similar findings, with a prevalence of symptoms of GAD in 69.6% and 73% of teachers, respectively. Ozamiz-Etxebarria et al. [42] found a prevalence of 49.5%, with severe and extremely severe symptoms in 7.6% of the teachers. About 21% of the teachers reported a moderate level of GAD. An American cross-sectional study showed a slightly lower prevalence (38.4%) in special education teachers. This value is twelve times higher than the prevalence of GAD observed in a survey of the general population at the beginning of the 21st century [17].

The development of GAD symptoms was associated with the teachers’ levels of concern. In a Chinese study, 20.21% of the very worried teachers showed moderate or high levels of anxiety, while the other categories (not worried, not so worried, generally worried, and more worried) showed some level of anxiety. Teachers with no or lower levels of anxiety were less likely to be very worried than those with high levels of anxiety [49]. A similar prevalence (23.1%) was observed in an American cross-sectional study [45].

Although most studies have shown, regardless of the instrument used, prevalence of the disorder at similar levels [50,51]. Analysis of GAD levels before and during the pandemic shows evidence of increased disturbance among teachers [10,13,25]. A study by Randall et al. [44] shows the preponderance of moderate-intensity GAD among teachers. The analysis of these symptoms during the different peaks of the pandemic in different regions and periods also showed that levels were higher in the second wave compared to the first [13,40].

Some characteristics were significantly associated with risk factors and important contributors to the onset of GAD symptoms, such as younger age, female gender, having children, black ethnicity, low income, presence of chronic illness, or depression before the pandemic, job stability for less than three months and being in the first five years of the profession, [13,21,24,26,31,38,41,42,46,50]. On the other hand, in Kukreti et al. [31], the type of school was not associated with a higher occurrence of GAD levels among teachers.

Concerning the teaching experience, high levels of fear of the COVID-19 pandemic and feelings of nomophobia were positively associated with higher levels of stress [31]. The comparison between white and black teachers reveals that common stressors were shared regardless of ethnicity, such as separation from family and close friends. However, other factors were not commonly shared, with black teachers experiencing significantly more protective factors than white teachers, including spending more time with family, paying more attention to personal health, and finding truer meaning at work [18].

Some investigated psychological states also contributed to triggering changes. The need for autonomy, competence, connection with relatives, and a sense of control were significantly associated [51], in addition to a low motivation for distance learning, and thoughts and feelings directly related to COVID-19, such as the need to practice social distancing and economic impacts [16]. In Chan et al. [20], GAD was positively and significantly associated with emotional exhaustion, but not with job satisfaction. Fear of falling ill partly explained the high levels of GAD [16,31]. The COVID-19 disease itself as a stressor was significantly associated with the onset of anxiety and depression [41].

### 3.2. Post-Traumatic Stress Disorder, Teaching and the COVID-19 Pandemic

In our review, only three studies evaluated the association of the coronavirus pandemic with the onset of post-traumatic stress disorder (PTSD) in teachers [22,31,47]. Although teachers are not considered a group of workers usually exposed to trauma, such as health workers and other categories, a cross-sectional study observed that PTSD symptoms above the severity threshold in teachers were not infrequent, with a prevalence of 29.1%, a result similar to that of patients with rheumatologic diseases, who usually have psychiatric disorders [47]. In Wuhan, the prevalence of PTSD among teachers was 24.55%, and it was even higher (46.81%) among teachers infected with COVID-19 [22].

Kukreti et al. [31] showed a lower proportion of post-COVID-19 PTSD among the population studied, with a prevalence of 12.66% and an association between the diseases mediated by psychological distress.

### 3.3. Burnout Syndrome, Teaching and the COVID-19 Pandemic

Studies that evaluated the occurrence of burnout syndrome (BS) [8,9,17,21,30,33,40,47] in teachers during the pandemic invariably showed some degree of association between the work context and the outbreak of the disorder, with a prevalence greater than 50% [8,17] and an increase in the prevalence ratio as the pandemic evolved [40]. The positive and negative predictors involved, although different, were somehow related to the stressful and exhausting work context, with a strong impact on the dimensions of the disease.

Analysis of these dimensions reveals that emotional exhaustion and depersonalization were high in 24% and 15.5% of the subjects in an African cross-sectional study, respectively, while the sense of professional accomplishment was low in 39% of the teachers [17]. Other findings also suggest an extreme impact of the COVID-19 pandemic on the feeling of BS by American special education teachers, with 86% of the subjects reporting the association [21]. All these dimensions showed a significant positive correlation with an intention to change jobs, with emotional exhaustion being the most important predictive factor for the intention to change professions, followed by depersonalization [33].

Resilience and its dimensions were aspects involved in the BS at work, having a significant negative correlation with the disorder and being able to significantly and negatively predict BS [33]. Job satisfaction was also a predictor factor, playing an important role in professional identification and BS at work [8]. Additionally, social aspects and changes in work practices specific to the context produced by COVID-19 were also associated with the development of the BS, such as the need to use and develop skills in new information and communication technologies, the workload increased by distance education, work/family conflict, social support, greater difficulties in dealing with stress and the inclusion of teachers in COVID-19 risk groups [17,30,40,47].

The investigation of BS dimensions reveals the centrality of BS and technostress (defined as unsuccessful adaptation due to failure in dealing with technology and changes in the requirements related to the use of technology, which generate psychological stress) in the outbreak of work-related disorders [9].

### 3.4. Depressive Disorder, Teaching and the COVID-19 Pandemic

Eight studies evaluated an association between the profession and the occurrence of depression in the context of the COVID-19 pandemic [13,21,23,25,26,27,29,31,32,34,35,41,42,44,45,50]. The prevalence of depressive symptoms ranged from 32.2% to 67% between studies. In a Turkish study [45] whose prevalence was 42.7%, the occurrence of depression was similar to that in a group of people with rheumatological diseases and significantly lower than the control group, composed of health workers. In this survey, 15.3% of the teachers had a history of psychiatric disorders or medication use in the last three months.

A similar prevalence was reported in the first wave of the pandemic in a Polish cohort, in which most teachers had moderate symptoms (16.55%), while 13.11% had severe or extremely severe symptoms [13]. In the same study, the prevalence of depression among teachers in the second wave was about 10% higher, with a significant increase in teachers with severe or extremely severe symptoms (22.86%) [13]. The higher prevalence were recorded by an American study—67% of teachers presented depression—and a Chinese study, with 56.9% of the subjects scoring a value greater than the cutoff point considered in the questionnaire used, with 16.1% manifesting moderate to severe depression [50].

On the other hand, the lowest values were recorded by a Spanish survey, with a prevalence of 32.9% and most of them manifesting mild symptoms (12.7%) [42], and an American survey, with a ratio of 37.5% [21]. In contrast to such expressive results, another American study showed an average that was lower than the cut-off point used for depression, demonstrating a lack of significance in the population studied [44].

The studies analyzed different risk factors and showed variability of significance in relation to those commonly analyzed. Two studies showed a significant association with gender [42,50], while one showed opposite results [31]. In Zhou et al. [50], gender was not a predictor for the occurrence of depressive disorder in the first wave of the pandemic, but it was in the second wave. Age as a predictor for depression also showed a variable and antagonistic result between studies [42,50].

The psychological state of professionals (such as the level of mental resilience, social, emotional and instrumental support, and fear of COVID-19), changes in social relationships with the spouse and family, and changes in work aspects (such as increased working time, need for distance learning, job instability) were significantly associated with the occurrence of depression [13,23,25,31,32,42,50]. Issues directly concerning the use of technology in distance learning were closely associated with the occurrence of distress in Navarro-Espinosa et al. [41].

High levels of stress were also associated with depressive disorder [25,31,50]. There was no association between depression and having children [13,42], and no association with years of experience and level of education [41]. A state of teacher affiliation with students was observed to be protective [25], in addition to significant teaching experience and less perceived overworking before and during COVID-19 pandemic [35].

## 4. Discussion

Although there is variability among the studies analyzed, the risk factors most correlated with increased psychological morbidity in teachers were: (i) being female; (ii) age below the fifth decade of life; (iii) pre-existence of chronic or psychiatric illnesses before the pandemic; (iv) difficulty in adapting to the distance education model; (v) family/work conflicts; (vi) negative symptoms caused by the pandemic.

Some of these factors have already been reported in the pre-pandemic period. The Brazilian cohort [52] demonstrated worse mental health in female teachers. The same findings are corroborated by UK and German cohorts. In the first case, associated with gender, the fact of living alone, the high workload and the pre-existing comorbidities (e.g., having a disability) were factors that worsened mental health in teachers [53]. In the second case, in addition to work overload, the authors point to the difficulty of disconnecting from work as an important factor in the worsening of the mental health of German teachers [54].

Kang et al. when analyzing 54 studies with 256,896 teachers in 22 countries, observed that anxiety, depression and stress are associated with various socio-demographic and institutional factors, including gender, nature of online teaching, job satisfaction, teaching experience, and the volume of workload. Additionally, several protective factors, such as regular exercises and provision of technical support for online teaching, reduced teachers’ negative psychological experiences [55].

It is important to note the variation in the incidences of GAD, BS, PTSD and depression in the literature, probably due to the lack of standardization of instruments among the surveys and the heterogeneity of the analyzed samples in terms of age, gender, and nationality [13,16,18,20,21,25,31,36,41,42,43,44,50,51,56,57]. Among the factors that most influenced these two variables are excessive workload [56], difficulty in managing/acquiring new skills associated with remote education [9], the existence of family/work conflicts [17,40], and impaired quality of life levels before/after the pandemic [15].

In the analysis of GAD symptoms, the female gender [57,58], fear of being infected with the disease, concerns about the course of the pandemic [59], and problems in school communication [60] concerning a smooth transition between face-to-face to remote teaching, leading to the deficient development of technologies that attract students’ attention and consequent students’ engagement [42,61] were the most correlated factors [13].

It is possible to infer that the occurrence of PTSD in teachers after the COVID-19 pandemic is multifactorial. PTSD seems to arise as an interaction between stressful situations of the pandemic: (i) fear of becoming infected/dying; (ii) fear that the same will happen to their family members and friends; (iii) lack of social support due to social distancing; (iv) stigmatization of the disease for those who became infected; and (v) new work demands arising from distance education—online classes, difficulty in separating personal/professional life, lack of adequate rest, distress regarding the expectations of having to learn technology skills in a timely manner (e.g., editing video, photos, recordings, etc.) [9,19,31,59]. However, it is important to note that our review presents limited results regarding the occurrence of PTSD in teachers due to the low number of studies that evaluated such correlation.

The Chinese cohort corroborates our findings. When evaluating over 80,000 professionals, there was a positive association between female gender, age over 60 years, living in larger cities, lower academic training, working in initial grades, unhealthy habits and higher levels of concern and fear for the pandemic with increased psychiatry disorders [57]. Similar findings also can be seen in two recent reviews [55,62].

In our review, almost half of the teachers experienced depressive symptoms, with rates similar to that of groups affected by chronic diseases such as rheumatologic patients. In parallel, severe depressive symptomatology occurred at rates greater than 10%. The risk factors mentioned in the literature are the same as those that lead to high levels of stress, as previously mentioned, except for gender, whose findings are discordant [13,21,25,31,41,42,44,45,48,50]. In another review, no significant gender differences were found for the occurrence of anxiety, stress levels, and depression either [63].

The reason for these variations may be due to particularities in the individual coping mechanisms for the pandemic. Cultural practices and the way in which individuals relate to each other and to the organizations to which they belong emerge in the literature as a new way of assessing the impact of COVID-19 on communities. This ultimately reflects the fact that people have reacted differently to COVID-19 around the world [64].

Our review indicates a higher prevalence of anxiety (38.4% to 69.6%), depression (32.2% to 54.99%) and stress (12.66% to 29.1%), than that observed in another review with a meta-analysis of studies published from December 1, 2019 to June 15, 2021, which show that teachers report levels of anxiety (17%), depression (19%), and stress (30%). Higher anxiety levels were found at elementary levels of education, but more teachers with stress were found among university professors [42]. Comparing their data to those observed in studies on the same disorders in the general population, during the pandemic, the authors observed lower prevalence’s. This strengthens the hypothesis that the aforementioned factors exerted pressure on the teachers and may have contributed to the development of the referred disorders.

In a scope review, which included 70 studies conducted on five continents, published from 1974 to 2022, the prevalence of BS ranged from 25.12% to 74%, stress ranged from 8.3% to 87.1%, anxiety ranged from 38% to 41.2% and depression ranged from 4% to 77%. An increase in stress was observed after the epidemic (from 2020 to 2022). The lowest, highest and median stress prevalence ranges were, respectively, 6%, 66% and 10.7%, while the prevalence up until 2019 (prior to the pandemic and lockdown) were, respectively, 7%, 100% and 33.9%. For burnout, the prevalence increased from 2.81%, 63.43% and 25.09% (prior to the pandemic and lockdown) to 3.1%, 70.9% and 27.6% (after the pandemic). Regarding the anxiety symptoms prevalence after the pandemic and lockdown were, respectively, 10.5%, 66.0% and 38.9%. Similarly, the lowest, highest, and median anxiety prevalence’s up until 2019 were, respectively, 7.0%, 68.0% and 26.0%. As for depression, the prevalence did not show a marked change between the referred periods, since until 2019 they were, 0.7%, 85% and 24.1% and after the pandemic and lockdown 0.6%, 85.7% and 23.5% [55].

Finally, as limitations of our review, the following can be highlighted: (i) non-homogeneity of the analyzed samples—teachers working at different levels of education (e.g., preschool, primary education, secondary education, university) and they are from several countries; (ii) non-standardization of the instruments used for screening/diagnosing the psychiatric pathologies analyzed; (iii) limited number of studies; (iv) the different study designs, with only four longitudinal studies in the sample, which makes it difficult to elaborate a concrete cause–effect. Therefore, the findings must be carefully evaluated.

## 5. Conclusions

High levels of GAD, burnout syndrome and depression were the most prevalent disorders among teachers in the context of the COVID-19 pandemic according to the literature. Studies about PTSD were scarce. Being female, high workload, low motivation to perform the function, and the changes typical of the period, such as changes in family and social relationships, fear of the disease, and social distancing are important risk factors to the aforementioned conditions.

Given the limited number of studies in this area, there is a need for more longitudinal studies comparing teachers in different countries and levels of education (e.g., preschool, primary education, secondary education, university) to obtain a broader understanding and draw more robust conclusions.

## Figures and Tables

**Figure 1 ijerph-20-01747-f001:**
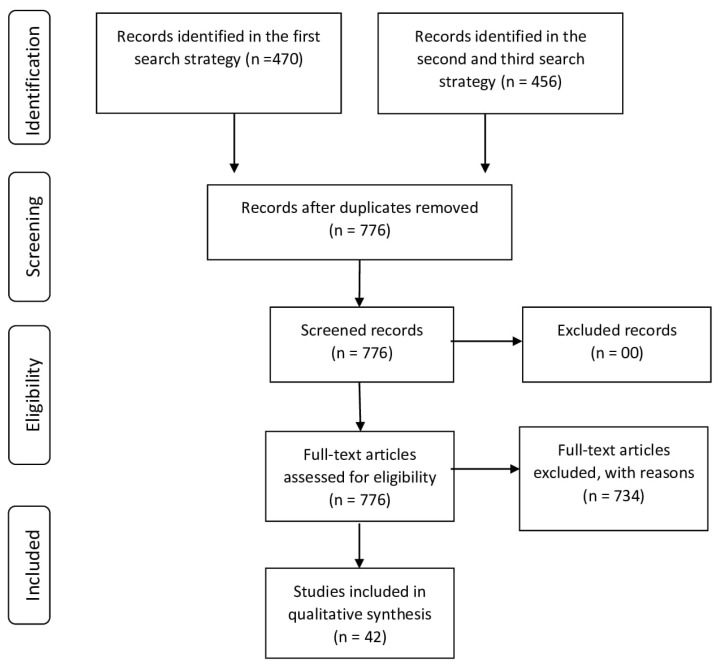
PRISMA flow diagram.

**Figure 2 ijerph-20-01747-f002:**
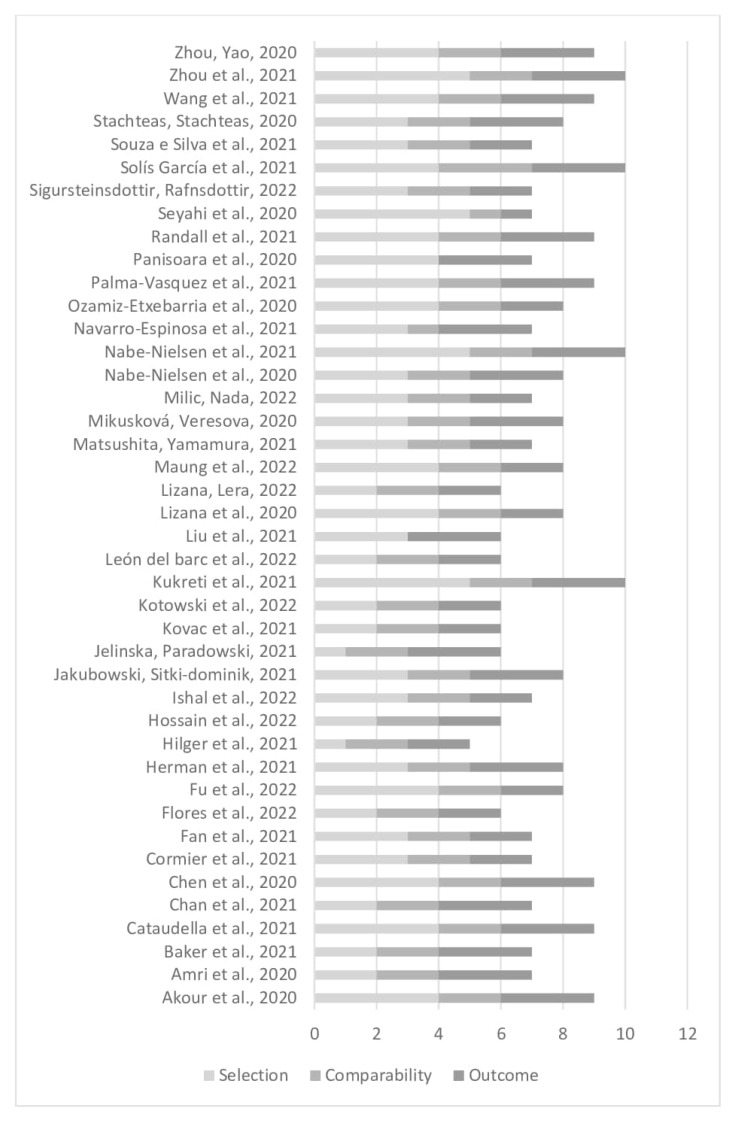
Quality assessment by the Newcastle Ottawa Scale [10,13,16,18,20,21,24,25,26,27,29,31,34,36,37,38,40,41,42,44,45,46,49,50,51].

**Table 1 ijerph-20-01747-t001:** Synthesis of the characteristics of the articles included in the systematic review.

Author (Year)-Quality of Studies	Study Type	Pathology Condition Assessed	Scales Used	Main Results	Study Limitations
Akour et al., 2020 [16] - Good	Cross-sectional	Generalized anxiety disorder (GAD)	Author questionnaire with a five-point Likert scale; Kessler Distress Scale (K10)	The younger you are, the more likely you are to develop generalized anxiety disorder (GAD). University professors with low motivation for distance learning are 1.42 times more likely to experience moderate to severe GAD than the reference category. There were 344 subjects (90.1%) reported not using any medication in response to the COVID-19-induced psychological impact, while 38 respondents (9.9%) reported using different medications and sedative–hypnotic drugs. Most subjects (69.6%) faced varying degrees of GAD amid the current pandemic.	Small and non-randomized sample; Self-report questionnaire; The sample was composed for teachers from more developed urban areas; Inability to establish causality by study design.
Amri et al., 2020 [17] - Good	Cross-sectional	Burnout Syndrome	Author questionnaire; Maslach Burnout Inventory	54% of the teachers had burnout. 38% experienced mild burnout, 12% moderate, and 6% severe.	Small and non-randomized sample; Self-report questionnaire; Impossibility of assessing causality.
Baker et al., 2021 [18] - Good	Cross-sectional	Generalized anxiety disorder (GAD)	Author questionnaire	As for their overall mental health since the beginning of the COVID-19 pandemic, most teachers reported that it was between “fair” and “good” (M = 2.84, SD = 1.05). Black teachers reported better mental health than white teachers (*p* < 0.001). Mental health was predictably inversely associated with coping and learning outcomes.	Self-report questionnaire; Use of non-validated questionnaire; Inability to establish causality by study design.
Cataudella et al., 2021 [19] - Good	Cross-sectional	Self-esteem; Self-efficacy	Rosenberg Self-Esteem Scale; Four-point Likert scale; Teacher’s Self-Efficacy Scale	The results showed lower self-esteem and lower self-efficacy on the part of teachers compared to the normative sample. They were also lower in teachers with more years of work. The teachers perceived a greater difficulty among the students than among them.	Small and non-randomized sample; Inability to establish causality by study design.
Chan et al., 2021 [20] - Good	Cross-sectional	Emotional exhaustion; Stress	Maslach Burnout Inventory Educators Survey; 5-point Likert scale; Teacher Stress Inventory job satisfaction subscale; Teacher’s Subjective Well-Being Questionnaire; Teaching Autonomy Scale; Revised Teacher Stress Inventory Subscales	A total of 50% of the respondents reported feeling emotionally challenged about their teaching experience frequently since school closing. A total of 59% of the subjects felt the stress of the daunting task more than ‘sometimes’. Over 51% of the respondents felt unclear about their job duties and expectations more than ‘sometimes’, and 22.3% of the respondents felt satisfied with their jobs less frequently than ‘sometimes’ during school closures. Task stress had a robust and positive association with emotional exhaustion, but not with job satisfaction. Role ambiguity was negatively associated with job satisfaction, but not significantly associated with emotional exhaustion. All three job resources showed significant and positive associations with job satisfaction. In addition, school connection was negatively associated with emotional exhaustion.	Non-representative sampling study; Restricted to a large center in the US; Inability to establish causality by study design.
Chen et al., 2020 [8] - Good	Cross-sectional	Burnout syndrome	Teacher Professional Identity Scale; Job Satisfaction Scale; Burnout Scale at Work	Teachers’ professional identity and job satisfaction are significantly negative predictors of burnout at work, with job satisfaction playing a moderating role between professional identity and burnout at work. Professional identity and job satisfaction are important factors that affect the work strain of university professors.	Use of online questionnaire for research; Small sample with only university teachers; Inability to establish causality by study design.
Cormier et al., 2021 [21] - Good	Cross-sectional	Generalized anxiety disorder (GAD); Depression; Anxiety; Emotional exhaustion; Depersonalization; Personal fulfillment Burnout syndrome	Maslach Burnout Inventory—Educators Survey (MBI-ES); 7-point Likert scale; Patient Health Questionnaire (PHQ-9); Generalized Anxiety Disorder Scale (GAD-7); Teacher-Specific Stress	A total of 38.4% met clinical criteria for generalized anxiety disorder, a rate 12.4 times higher than that of the US population, and 37.6% for major depressive disorder, a rate 5.6 times higher than that of the population. The impact of the pandemic was moderate to severe on stress (91%), depression (58%), anxiety (76%), and emotional exhaustion (83%).	Use of online and self-report questionnaire for research; Inability to establish causality by study design.
Fan et al., 2021 [22] - Good	Cross-sectional	PTSD; Anxiety; Intrusion; Avoidance; hyperarousal	Author questionnaire; Impact of Event Scale-Revised for PTSD symptoms	The results showed that the overall incidence of post-traumatic stress disorder among college professors was as high as 24.55%, but the mean level of PTSD scores was low. Compared with those without symptoms, the proportion of PTSD increased by 181%. For those who had family members or relatives who died from COVID-19, the proportion was 459% higher than those who had no loss. The results showed that approximately 1/4 of university professors in Wuhan developed PTSD, but the level of PTSD was low	Addresses only one class of teachers Use of online and self-report questionnaire for research; Inability to establish causality by study design.
Flores et al., 2022 [23] - Fair	Cross-sectional	Depression; Anxiety; Stress; Satisfaction with life; Concerns; Social relationships; Economic situation; Work and family life	Escala de Diener, Emmons, Larsen e Griffin Brief Multidimensional Student Satisfaction with Life Satisfaction Scale DASS-21 Escala Breve de Espiritualidade	Spirituality and the use of self-applied socio-emotional strategies by teachers were not directly related to their mental health, so their mediating effect in relation to life satisfaction was discarded. Teachers who used social-emotional strategies, as well as those who reported higher levels of spirituality, had greater satisfaction with life, both generally and specifically. Women showed higher levels of symptoms of depression, anxiety and stress, but also higher levels of satisfaction with life.	Non-randomized sample; Use of online and self-report questionnaire for research; Inability to establish causality by study design.
Fu et al., 2022 [24] - Good	Cross-sectional	GAD	Generalized Anxiety Disorder Scale (GAD-7); Multidimensional Perceived Social Support Scale (MSPSS)	One year after the start of the COVID-19 pandemic, the overall prevalence of anxiety was 40.0%, being higher in women than in men. Being female, age ≥60 years, being married, and poor family economic status were significantly associated with anxiety. Participants with moderate, mild, or no impact of COVID-19 on their lives showed a reduced risk of anxiety compared to those who reported a significant effect. Anxiety symptoms were found in about two-fifths of Chinese university professors 1 year after the start of the COVID-19 pandemic.	Use of online and self-report questionnaire for research; Only public universities; Inability to establish causality by study design.
Herman et al., 2021 [25] - Good	Cross-sectional	Generalized anxiety disorder (GAD) Depression	General health and job satisfaction scales; Patient Health Questionnaire-2 (PHQ-2); Generalized Anxiety Disorder-2 (GAD-2); Authoritative School Climate Survey (ASCS)	Teachers reported job stress levels above the midpoint, indicating that their jobs were more stressful, and they also reported values showing high levels of job stress. The mean values of teachers’ job satisfaction and general health status were positive. As for depression and GAD, 9% of teachers reported scores above the cutoff on the PHQ-2, indicating that major depression was likely, and 16% of the teachers reported scores above the cutoff on the GAD-2, indicating a probable risk for GAD.	Possibility of memory bias; Impossibility of assessing causality.
Hilger et al., 2021 [14] - Good	Longitudinal research	Autonomy; Social support; Fatigue; Claims; Job satisfaction; Stress; Affectivity	Author questionnaire with a 7-point Likert scale; Copenhagen Psychosocial Questionnaire; Subscale of Emotional Work Demands; Quantitative Workload Inventory; Scale of Interpersonal Conflict at Work; Work Demand Questionnaire; Three-Dimensional Work Fatigue Inventory Freiburg; Bodily Complaints Inventory	Both work demands and work resources have decreased compared to the period prior to COVID-19. Furthermore, we found simultaneous changes in work-related well-being in terms of decreased fatigue. No overall changes were found for psychosomatic symptoms, complaints, and job satisfaction. Work demands and work resources differentially impacted well-being, while changes in work demands—but not work resources—were related to negative well-being (fatigue and psychosomatic complaints). Changes in work resources, but not work demands, were associated with changes in positive indicators of well-being (job satisfaction).	Original study design was not for assessing the impact of the covid-19 pandemic; Use of online and Self-report questionnaire for research; Possibility of memory bias.
Hossain et al., 2022 [26] - Good	Cross-sectional	Depression; Anxiety; Stress; Fear	DASS-21 Author questionnaire	The results indicate that the overall prevalence of depression, anxiety and stress among teachers was 35.4%, 43.7% and 6.6%, respectively. Prevalence was higher among male and older teachers than among female and younger colleagues. The results also showed that the place of residence, the institution, self-reported health, use of social and electronic media, and fear of COVID-19 significantly influenced the mental health status of teachers. University professors showed less depression compared to primary school professors. The results showed that teachers aged 31–40 years old and 41 years old or older were less stressed compared to teachers younger than 30 years old.	Small and non-randomized sample; Use of online and self-report questionnaire for research; Inability to establish causality by study design.
Ishak et al., 2022 [27] - Good	Cross-sectional	Depression; Anxiety; Stress; Quality of life; Dysphoria; hopelessness; Self-depreciation; Nervousness; Difficulty sleeping; physical functioning	DASS-21; SF-36 Health Survey Questionnaire (SF-36); Author questionnaire	According to the findings of this study, most teachers (55.5%) were anxious, followed by depression (39.9%) and stress (27.6%). Depression, anxiety and stress were all statistically related to age, marital status and number of children. When it comes to quality of life, teachers had the highest physical functioning score at around 86 but the lowest vitality at 62.3. All domains of quality of life were negatively correlated with depression, anxiety and stress.	Small and non-randomized sample; Use of online and self-report questionnaire for research; Only teachers from primary and secondary schools; Inability to establish causality by study design.
Jakubowski, Sitko-Dominik, 2021 [13] - Good	Retrospective research	Generalized anxiety disorder (GAD); Depression	The Depression Anxiety & Stress Scales-21 (DASS-21); The Berlin Social Support Scales (BSSS); The Relationship Satisfaction Scale (RS-10); The Injustice Experience Questionnaire (IEQ)	The disorders were more prevalent in the second wave of the pandemic in Poland. Stress ranged from 6% in the first stage of the survey to 47% in the second stage. Anxiety and depression ranged from 21 to 31% and from 12% to 46% between waves, respectively.	The first phase of the study was retrospective and the second was cross-sectional; Possibility of memory bias; Non-homogeneity in the sample analyzed between the first and second steps.
Jelińska, Paradowski, 2021 [28] - Good	Cross-sectional	Sadness; Irritation; Willingness; Emotional instability; Fatigue; Loneliness	23 short scales were developed from the International Personality Item Pool [IPIP]; Six-point Likert scale; Perceived Stress Scale (PSS); Self-Compassion Scale	Negative affect was significantly and positively correlated with greater situational generalized anxiety disorder (GAD) (r = 0.47) and situational loneliness (r = 0.36). Furthermore, the stronger the negative emotional state, the less teachers reported work–life synergy (r = –0.43) and the less productive they felt (r = –0.33) during the pandemic. Finally, those who felt more negative emotions were also coping worse than others (r = –0.30). The most influential predictor of teachers’ negative emotional states was anxiety (*p* < 0.001), explaining approximately 22% of the variation in negative affect.	Self-selection biases of participants: use of an online questionnaire (deprives those who do not have access to the internet), selects those who would be motivated to participate or identify with the theme; Inability to establish causality by study design.
Kovac et al., 2021 [29] - Good	Cross-sectional	Depression; Anxiety; Stress	DASS-21	High prevalence of high levels of depression (31.5%), anxiety (39%) and stress (19%). Teachers have a high prevalence of high levels of depression, anxiety and stress. Teachers have a higher level of depression, anxiety and stress than male teachers. There was no difference in depression, anxiety, and stress between elementary school teachers and high school teachers. Protective factors for depression, anxiety, and stress are levels of family support, school administration support, and student understanding. Depression, anxiety and stress levels were not statistically different in vaccinated and unvaccinated teachers.	Non-randomized and not representative sample; Only elementary school teachers and high school teachers; Inability to establish causality by study design.
Kotowski et al., 2022 [30] - Good	Cross-sectional	Stress; Exhaust; Welfare; Burnout	Author questionnaire	Stress and burnout remain high for teachers, with 72% of teachers feeling very or extremely stressed and 57% feeling very or extremely drained. Many teachers struggle to have a satisfactory work–family balance (37% never or almost never; 20% only sometimes)	Inability to establish causality by study design; Sample selected for convenience; Use of a self-administered and online questionnaire.
Kukreti et al., 2021 [31] - Good	Cross-sectional	Post-traumatic stress disorder; Nomophobia; Fear; Psychological suffering; Depression; Generalized anxiety disorder (GAD)	PTSD Checklist for DSM-5 (PCL-5); Fear of COVID-19 PTSD Scale; Chinese COVID-19 Fear Scale (FCV-19S) of seven items; 20-item Chinese Nomophobia Questionnaire; Chinese Depression, Anxiety and Stress Scale (DASS-21)	With the cutoff score of 31, the prevalence of PTSD was 12.3%, but decreased to 1.0% when the cutoff score was 49. Fear of COVID-19 among teachers leads to PTSD due to psychological distress, highlighting the moderating effect of nomophobia in this association.	Inability to establish causality by study design; Sample selected for convenience; Use of a self-administered and online questionnaire.
Koestner et al., 2022 [32] - Good	Cross-sectional	Emotional exhaustion; Depersonalization; Anxiety; Depression	Validated German version of the Patient Health Questionnaire 4; Maslach Burnout Inventory (MBI); Author questionnaire	The psychological burden on German teachers exceeded the level of the general population, for example, in relation to symptoms of depression or generalized anxiety. Subgroup analysis revealed that psychological burdens were unequally distributed among different groups of teachers; Younger teachers (18–30 years old) had more symptoms of depression compared to their older colleagues (56–67 years old)	Inability to establish causality by study design; Sample selected for convenience; Use of a self-administered and online questionnaire.
Liu et al., 2021 [33] - Good	Cross-sectional	Resilience; Professional exhaustion; Intention of turnover; Burnout; Confidence; Optimism; Force	Resilience Scale developed by Connor and Davidson; Work exhaustion scale; Rotation intention scale	Burnout had a significant positive predictive effect on turnover intention (r = 0.485, *p* < 0.05). At the same time, job burnout played a moderating role in resilience and turnover intention (*p* < 0.001). Resilience and its dimensions of confidence, strength, and optimism had a significant negative correlation with turnover intent, and resilience could significantly predict turnover intent negatively.	Sample restricted to one region; Use of non-validated scales for the purpose of the study; Use of a self-administered and online questionnaire for the government system.
Lizana et al., 2021 [15] - Fair	Longitudinal research	Feelings (uncertainty, loneliness, fear, role limitations due to emotional problems and mental health)	Short-Form 36 Health Survey (SF-36) questionnaire.	Quality of life showed a significant decrease during the pandemic compared to the pre-pandemic measure (*p* < 0.01). In each gender, there were significant differences between the pre-pandemic and pandemic times, with a greater impact among women in the summary variables of the mental and physical component and in seven of the eight quality of life scales (*p* < 0.01). Across age categories, people under 45 showed significant differences between pre-pandemic and pandemic times across all summary dimensions and measures.	Use of online and Self-report questionnaire for research; Sample size.
Lizana, Lera, 2022 [34] - Good	Cross-sectional	Depression; GAD; Stress	DASS-21	High rates of symptoms of depression, anxiety and stress were observed among teachers (67%, 73% and 86%, respectively). Among teachers affected by work–family balance (89%), there was also an increased risk of symptoms of anxiety (OR: 3.2) and stress (OR: 3.5). The risk of depression symptoms was higher among females (OR: 2.2), and teachers under 35 years of age were at risk of having all three symptoms (depression OR: 2.2; anxiety OR: 4.0; stress OR 3.0).	Small and non-randomized sample; Only teachers from primary and secondary schools; Use of online and self-report questionnaire for research; Inability to establish causality by study design.
Maung et al., 2022 [35] - Good	Cross-sectional	Depression; GAD; Stress	DASS-21; Author questionnaire	The percentages of respondents with mild, moderate, severe, and extremely severe depression were 12%, 9.7%, 4.7%, and 3.1%, respectively. Those with mild, moderate, severe, and extremely severe anxiety accounted for 11.5%, 12.3%, 6.3%, and 6%, respectively. Those with mild, moderate, severe and very severe stress accounted for 12.8%, 12%, 5.3% and 20.5%, respectively. Perceived overwork was significantly higher during the pandemic compared to before the pandemic. Significant teaching experience and less perceived overwork before and during the pandemic were associated with better mental health.	Only primary and secondary school teachers; Small and non-randomized sample; Use of online and self-report questionnaire for research; Inability to establish causality by study design.
Matsushita, Yamamura, 2021 [36] - Good	Cross-sectional	Generalized anxiety disorder (GAD) Reliability; Irritability; Fatigue	Brief Job Stress; Author questionnaire	A total of 59.6% of the subjects worked 11 h or more a day, with significant differences in workload between positions. Regarding effective teachers, gender (female), age, class tenure, number of years working in the same school, working hours of more than 10 h were significantly associated with high stress, compared to those who worked less than 9 h per day.	Non-randomized sample; Addresses only one class of teachers Use of online and self-report questionnaire for research; Impossibility to deduce causality of the relationship between long working hours and stress responses.
Mikušková, Verešová 2020 [37] - Good	Cross-sectional	Generalized anxiety disorder (GAD) Satisfaction; Expectation	6 point scale; Positive and Negative affect Schedule (PANAS); Big Five Inventory (BFI-2)	During the pandemic period, teachers’ negative emotions increased, while positive ones decreased; distance education was closely related to emotions (and changes in emotions) and personality; In addition, teachers reported willingness to implement partial changes in their teaching after the pandemic period.	Small and non-randomized sample; Use of online and self-report questionnaire for research; Only primary school and upper-secondary school teachers; Inability to establish causality by study design.
Milić, Marić, 2022 [38] - Good	Cross-sectional	GAD	Author questionnaire; GAD-7	On the GAD-7 scale, the arithmetic mean of the examined population is 0.64 (SD 1.22), which indicates a normal degree of anxiety in the examined population. Only 2% of teachers had mild anxiety. Moderate and severe anxiety were not reported.	Small sample, non-randomized; Only primary school and upper-secondary school teachers; Inability to establish causality by study design.
Nabe-Nielsen et al., 2020 [39] - Good	Cross-sectional	Fear	Author questionnaire	The prevalence of concern about going to work was higher among those who were solely responsible for remote teaching (34%) than among those who taught at school (19%). Other emotional reactions, that is, fear of contagion and transmission of infections, were equally frequent among those who carried out face-to-face teaching at school and those who carried out online remote teaching.	Addresses only public school teachers; Use of an online, non-validated and self-administered questionnaire for research. Inability to establish causality by study design.
Nabe-Nielsen et al., 2021 [40] - Fair	Longitudinal research	Generalized anxiety disorder (GAD) Exhaustion; Fear; Burnout syndrome	Author questionnaire	Emotional reactions and poor mental health increased significantly from 27 to 84% from May to November and December 2020. Teachers, particularly vulnerable to the adverse consequences of COVID-19 had the highest prevalence of fear of infection and poor mental health. Teachers who reported being part of a COVID-19 risk group had a higher level of emotional reactions and poor mental health at most measurement points.	Addresses only public school teachers; Use of an online, non-validated and self-administered questionnaire for research.
Navarro-Espinosa et al., 2021 [41] - Good	Cross-sectional	Generalized anxiety disorder (GAD) Depression; Technostress	Anxiety and Risk of Depression Scales; Rosenberg’s subscales of anxiety and depression risk	The risk of developing anxiety among teachers was 85.5%, with a similar frequency between the Spaniards (83.3%) and Ecuadorians (86.5%). The risk of developing anxiety showed significant differences in the role of Information and Communication Technologies (ICTs) in education and the lack of ICTs (*p* < 0.05). In addition, COVID-19 as a stressor showed a significant difference between teachers at risk of anxiety (82.98%), while the risk of depression was related to COVID-19 as a stressor (*p* = 0.037), the balance between family and work (*p* = 0.006), availability of computers and internet, role of ICT in education, lack of models and time (*p* < 0.05). 85% of the college professors were at risk of developing anxiety, which was also linked to depression.	Small sample; Use of online and self-report questionnaire for research; Only university teachers and specific knowledge are; Inability to establish causality by study design.
Ozamiz-Etxebarria et al., 2020 [42] - Good	Cross-sectional	Generalized anxiety disorder (GAD) Anxiety; Depression	Depression, Anxiety and Stress Scale-21 (DASS-21)	A total of 50.6% of the teachers indicated suffering from stress, with 4.5% reporting extremely intense stress and 14.1% severe stress. About 49.5% of the teachers reported suffering from anxiety, of which 8.1% reported extremely severe symptoms and 7.6% severe symptoms. Finally, 32.2% of the teachers reported suffering from depression, of which 3.2% reported extremely severe symptoms and 4.3% severe symptoms.	Self-report questionnaire for research; Inability to establish causality by study design.
Palma-Vasquez, Carrasco, Hernando-Rodriguez, 2021 [43] - Good	Cross-sectional	Depression; Generalized anxiety disorder (GAD) Diverse feelings	General Health Questionnaire (GHQ-12); Likert Scale; Score scale corrected using Cronbach’s alpha	The prevalence of poor mental health among teachers was 58.27%. The risk of having worse mental health was higher for those who worked two or more unpaid overtime hours a day compared to those who did not have to extend their work hours during the pandemic. Being absent due to illness was associated with a 282% greater likelihood of having mental health issues compared to not being absent during 2020.	Small, non-randomized sample; Only teachers who teleworked more than 50% during the 2020 academic year; Use of online and self-report questionnaire for research; Inability to establish causality by study design.
Panisoara et al., 2020 [9] - Good	Cross-sectional	Intrinsic and extrinsic motivation; Occupational stress (Burnout and technostress); Self-efficacy, burnout	Author questionnaire with a seven-point Likert scale; Work Tasks Motivation Scale for Teachers (WTMST); Oldenburg Burnout Inventory (OBI); Learning Misfit Scale; Person-Technology Enhanced (P-TEL); Continuation Intent Scale (CI)	The structural model accounted for 70% of the variation in teachers’ intention to stay using online teaching and for 51% of the variation in teachers’ burnout and technostress. Extrinsic motivation significantly amplifies occupational stress, represented by negative feelings about online teaching, while intrinsic motivation significantly diminishes them, even if with less intensity.	Use of online and self-report questionnaire for research; The experimental part of the study was carried out in a relatively short time frame; Inability to establish causality by study design.
Randall et al., 2021 [44] - Good	Cross-sectional	Physical well-being; Psychological well-being; Professional well-being; Work demands and life satisfaction during the pandemic; Satisfaction with life; depressive symptoms; Stress resilience; Secondary trauma	International Physical Activity Questionnaire (IPAQ); Short Last 7 Days Self-administered format; SF-12 Health Survey Standard, Version 1; Musculoskeletal Disorders Scale; Related to Work (WMDS); Life Satisfaction Scale; Center for the Epidemiological Studies of Depression 10-item Short Form (CES-D-10); Perceived Stress Scale (PSS); Brief Resilience Scale; Professional Quality of Life Scale subscales; Early Childhood Job Satisfaction Survey (ECJSS); Job Content Questionnaire (JCQ); Author questionnaire Hospital Anxiety and Depression Scale (HADS); Impact of Event Scale-Revised (IES-R)	Of the respondents, 77% were overweight or obese and only 39% met the recommendation of 150 min of moderate physical activity per week. They had an average life satisfaction score that qualifies as mild satisfaction, experienced moderate stress, and were collectively approaching the threshold for depression, but still reflected moderate to high work commitment.	Only early care and education teachers; Non-randomized sample; Use of online and self-report questionnaire for research; Inability to establish causality by study design.
Seyahi et al., 2020 [45] -Good	Cross-sectional	Generalized anxiety disorder (GAD) Depression; Post-Traumatic Stress; Sleep problems	International Physical Activity Questionnaire (IPAQ); Short Last 7 Days Self-administered format; SF-12 Health Survey Standard, Version 1; Work-Related Musculoskeletal Disorders (WMDs); Life Satisfaction Scale; Center for the Epidemiological Studies of Depression 10-item Short Form (CES-D-10); Perceived Stress Scale (PSS); Brief Resilience Scale; Professional Quality of Life Scale subscales; Early Childhood Job Satisfaction Survey (ECJSS); Job Content Questionnaire (JCQ); Author questionnaire Hospital Anxiety and Depression Scale (HADS); Impact of Event Scale-Revised (IES-R)	A total of 23.1% of the teachers had generalized anxiety disorder (GAD), 47.7% depression, and 29.1% post-traumatic stress symptoms.	Non-randomized sample; Selected group (patients with rheumatic diseases) Use of online and self-report questionnaire for research; Inability to establish causality by study design.
Sigursteinsdottir, Rafnsdottir, 2022 [46] - Fair	Longitudinal	GAD; Physical symptoms of tiredness; Sadness	10-item Perceived Stress Scale (PSS-10) Author questionnaire	Results show increased stress, worse mental and physical health, and increased mental and physical symptoms in 2021 compared to 2019. Worse mental health and mental health symptoms were the strongest predictors of high levels of stress. The interaction between time of study and mental health symptoms indicated that the influence of time was greater for teachers who reported mental health symptoms than for others. In other words, the COVID-19 pandemic was negatively correlated with these health and well-being factors, indicating the deterioration of teachers’ health and well-being. The proportional stress level increased by about 10% in 2021 compared to 2019, and the result revealed that 28% of teachers were categorized with high stress.	Only primary school teachers; Use of online and self-report questionnaire for research; Without confirmation by physicians.
Solís García et al., 2021 [47] - Good	Cross-sectional	Fatigue; Generalized anxiety disorder (GAD) Skepticism; Ineffectiveness	Survey Work-Home Interaction-Nijmegen (SWING); RED-ICT Technostress Questionnaire	Negative work–family interaction correlated with ineffectiveness (r = 0.12), skepticism (r = 0.09), fatigue (r = 0.96), and anxiety (r = 0.47). On the other hand, the family-work interaction was significantly correlated with the variables of ineffectiveness (r = 0.20), skepticism (r = 0.14), fatigue (r = 0.47), and anxiety (r = 1).	Only pre-school, primary, and secondary school teacher; Use of online and self-report questionnaire for research; Inability to establish causality by study design.
Souza e Silva e tal, 2021 [10] - Good	Cross-sectional	Insatisfação com o trabalho; Ansiedade; Depressão; Problemas de sono; Aumento do consumo de álcool; Medo da COVID-19	Author questionnaire	A total of 25.9% of teachers reported a formal diagnosis of anxiety and/or depression during the COVID-19 pandemic. During the pandemic, 7.1% of the teachers were drinking more alcohol than usual, 33.4% started having sleep problems, 30.4% were using drugs to relax/sleep/anxiety/depression, the perception of quality of life of 67.1% of teachers worsened and 43.7% reported having severe fear of COVID-19. Up to 82.3% of teachers had at least one condition related to mental health during the pandemic, such as increased alcohol consumption, sleep problems, use of psychotropic medication, quality of life, and fear of COVID-19. Most of the teachers reported a decrease in their quality of life during the pandemic. More than a quarter of the teachers in our study reported having received a formal diagnosis for anxiety and/or depression during the pandemic.	Only public school teacher; Use of online and self-report questionnaire for research; Self-report response, leading to the possibility of memory bias; Inability to establish causality by study design.
Stachteas, Stachteas, 2020 [48] - Good	Cross-sectional	Fear, Generalized anxiety disorder (GAD), optimism about the outcome, depression, desire to return to work, concern about the implementation of distance learning.	Author questionnaire with a 6-point Likert scale	A total of 34% of the teachers experienced a high and very high level of fear during the pandemic. The existence of an underage child in the family affects the experience of fear concerning the development of the pandemic. There is a correlation between gender and the emergence of feelings of fear, depression, and optimism. Women showed the highest levels of fear and depression, while the situation is the opposite when it comes to optimism. The optimism expressed also depends on education. Most teachers (60.6%) are moderately optimistic. Depression was correlated with sex, prevailing in lower-class men (69.9%).	Study included only/secondary school teachers, which makes it impossible to generalize; Small sample; Use of an online, non-validated and self-administered questionnaire for research.
Wang et al., 2021 [49] - Good	Cross-sectional	Attention level; Generalized anxiety disorder (GAD) Fear level; Behavior status to alleviate worry	Author questionnaire with a five-item Likert scale Generalized Anxiety Disorder (GAD-7)	A total of 59% of the teachers reported being ‘very worried’ about the pandemic. The proportion of female professors was higher than that of male professors (60.33% vs. 52.89%). In all age groups, the condition “very worried” represented the highest proportion. The 40–49 age group had the lowest proportion of very concerned subjects.	Use of online and self-report questionnaire for research; Inability to establish causality by study design.
Zhou et al., 2021 [50] - Good	Cross-sectional	Mental resilience Generalized anxiety disorder (GAD) Depression Sleep duration	Patient Health Questionnaire-9 (PHQ-9); Connor-Davidson Resilience Scale 25 (CD-RISC 25); Perceived Stress Scale-10 (PSS-10)	A total of 624 (56.9%) suffered from depression and 177 (16.1%) suffered from moderate to severe depression; 17 (10.7%) teachers had low mental resilience and 1053 (96.1%) teachers had a high perception of generalized anxiety disorder (GAD).	Use of online and self-report questionnaire for research; Inability to establish causality by study design; Study did not take into account important factors that influence depression, such as presence/absence of social support and physical activity.
Zhou, Yao, 2020 [51] - Good	Cross-sectional	Psychological states; Acute Stress Disorder	Received Social Support Questionnaire; Sheldon and Niemiec’s Basic Psychological Needs Scale; Subscale Sense of Control on the Feelings of Security Scale; DSM-5; Acute Stress Disorder Diagnostic Criteria B	A total of 68 (9.1%) subjects were identified as having probable ASD. The prevalence of acute stress symptoms in teachers is 9.1%. Social support had a negative indirect relationship with the Social support and acute stress symptoms through the need for autonomy or relationship, but not the need for competence.	Study included only primary/secondary school teachers, which makes it impossible to generalize; Inability to establish causality by study design; Assessment of only one group of factors (psychosocial) in the outcome.

**Table 2 ijerph-20-01747-t002:** Characterization of the samples and additional results of the studies selected.

Author (Year) - Country - Total Sample (*n*)	Female (*n*, %) - Age (Mean, Age Group or %)	Category of Teachers - Teaching Experience (Years)	Generalized Anxiety Disorder (*n*, % or M and SD)	Post- Traumatic Stress Disorder (*n*, % or M and SD)	Depressive Disorder (*n*, % or M and SD)	Burnout Syndrome (*n*, % or M and SD)	Psychiatric Illnesses before COVID-19 (*n*, %)	Work-Family Conflicts (*n*, %) - Family Support (*n*, %) - Social Support (*n*, %)
Akour et al., 2020 [16] - Jordan - 383	212 (55%) - 34–52 y	High - 1–53 y	120 (31.7%)	NA	NA	NA	14 (3.7%)	NA - 240 (62.8%) - 190 (49.7%)
Amrir et al., 2020 [17] - Morocco - 125	71 (65.8%) - 25–59 y	Elementary - 5–21 y	NA	NA	NA	67 (54%)	NA	70 (56%) - NA - 23 (18.4%)
Baker et al., 2021 [18] - US - 454	366 (80.6%) - 18–64 y	Elementary, Middle and High - 1–20 y	NA	NA	NA	NA	NA	NA - NA - 68 (15%)
Cataudella et al., 2021 [19] - Italy - 226	199 (88.1%) - 35–55 y	Elementary and High - 15 y	NA	NA	NA	NA	NA	NA - NA - NA
Chan et al., 2021 [20] - US - 151	120 (80.8%) - NA	Elementary - 2–46 y	89 (59%)	NA	NA	75 (50%)	NA	NA - NA - NA
Chen et al., 2020 [8] - China - 483	NA - NA	High - NA	NA	NA	NA	253 (52.4%)	NA	NA - NA - NA
Cormier et al., 2021 [21] - US - 468	415 (88.3%) - 43 y	Elementary and High - 32–54 y	178 (38%)	NA	275 (37.3%)	388 (83%)	NA	NA - NA - NA
Fan et al., 2021 [22] - China - 1650	855 (51.8%) - 32–48 y	High - NA	NA	24.55%	NA	NA	NA	NA - NA - NA
Flores et al., 2022 [23] - Chile - 624	464 (74.4%) - 33–55 y	NA - NA	NA	NA	NA	NA	NA	NA - NA - NA
Fu et al., 2022 [24] - China - 10,302	5760 (41.3%) - 18–60 y	High - 1–30 y	2380 (40%)	NA	NA	NA	NA	NA - 6761 (65.6%) - 2976 (28.8%)
Herman et al., 2021 [25] - US - 639	492 (77%) - NA	Elementary, Middle and High - 1–10 y	100 (16%)	NA	57 (9%)	NA	NA	NA - NA - NA
Hilger et al., 2021 [14] - German - 207	175 (85%) - 24–66 y	Elementary and Middle - 1–36 y	NA	NA	NA	NA	NA	NA - NA - NA
Hossain et al., 2022 [26] - Bangladesh - 381	149 (39.1%) - 30–40 y	Elementary, Middle and High - NA	43.7%	NA	35.4%	NA	NA	NA - 336 (88.2%) - NA
Ishak et al., 2022 [27] - Malaysia - 391	290 (74.2%) - 21–60 y	Elementary and Middle - NA	156 (55.5%)	NA	217 (39.9%)	NA	NA	NA - 232 (59.3%) - NA
Jakubowski, Sitko-Dominik, 2021 [13] - Poland - 285	Primary school, 130 (89.7%); Secondary school, 121 (86.4%) - 46.76 (primary); 43.76 (secondary)	Elementary and Middle - 19.07	21% in the first stage of the research to 31% in the second stage	6% in the first stage of the research to 47% in the second stage	12% in the first stage of the research to 46% in the second stage	NA	NA	NA - NA - Negative correlation (stress, anxiety and depression)
Jelińska, Paradowski, 2021 [28] - 92 countries - 804	578 (71.89%) - 44.1	High - NA	NA	NA	NA	NA	NA	NA
Kovac et al., 2021 [29] - Bosnia and Herzegovina - 559	471 (84.2%) - 59 (20–30 y); 211 (31–40 y); 184 (41–50 y); 105 (≥50 y)	Elementary and High - NA	39%	19%	31.5%	NA	NA	More family support is related to lower levels of stress, depression and anxiety.
Kotowski et al., 2022 [30] - Greater Cincinnati, Ohio, USA - 703	578 (82.2%) - 44.6 y	Elementary, Middle and High - 18.5 y	NA	72%	NA	57%	NA	Many teachers struggled to have a satisfactory work–family balance (37% never or almost never; 20% only has sometimes)
Kukreti et al., 2021 [31] - China - 2603	1865 (71.6%) - NA	Elementary and Middle - 10 or fewer years (56.2%)	NA	NA	321 (12.3%)	NA	NA	NA
Koestner et al., 2022 [32] - Germany - 31,089	24,099 (77.5%) - 45.8 y	Elementary, Middle and High - NA	M = 2.59 SD = 1.89	NA	M = 2.34 SD = 1.65	NA	NA	NA
Liu et al., 2021 [33] - China - 449	331 (73.72%) - 36.7 y	High - NA	NA	NA	NA	(M = 26.88, SD = 6.89)	NA	NA
Lizana et al., 2021 [15] - Chile - 63	45 (71%) - Male = 42.6 y Female = 38.9 y	Elementary and Middle - NA	NA	NA	NA	NA	NA	NA - NA - NA
Lizana, Lera, 2022 [34] - Chile - 313	256 (81.79%) - 37.9 y	Elementary and Middle - NA	230 (73.4%)	270 (86.2%)	209 (66.7%)	NA	NA	279 (89.14%) - NA - NA
Maung et al., 2022 [35] - Malaysia - 382	318 (83.2%) - 18–Older than 60 y	Elementary and Middle - 14.2 y	Mild: 44 (11.5%) Moderate: 47 (12.3%) Severe: 24 (6.3%) Extremely Severe: 23 (6%)	Mild: 49 (12.8%) Moderate: 46 (12%) Severe: 20 (5.2%) Extremely Severe: 8 (2.1%)	Mild: 46 (12%) Moderate: 37 (9.7%) Severe: 18 (4.7%) Extremely Severe: 12 (3.1%)	NA	NA	NA - NA - NA
Matsushita, Yamamura, 2021 [36] - Japan - 54,772	23,504 (42.9%) - ≤29 y ≥60 y	Elementary - <3 y (65%) ≥3 y (23,9%)	NA	NA	NA	NA	NA	NA - NA - NA
Mikušková, Verešová 2020 [37] - Slovakia - 379	340 (89.7%) - 45.14 y	Elementary and Middle - Primary school (19.56 y); upper-secondary school (17.10 y)	NA	NA	NA	NA	NA	NA - NA - NA
Nabe-Nielsen et al., 2021 [40] - Denmark - 871	675 (78%) - 50–59 y	Elementary, Middle and Secondary - NA	NA	NA	NA	NA	NA	NA - NA - NA
Navarro-Espinosa et al., 2021 [41] - Spain and Ecuador - 55	NA - NA	High - <10 y	NA	NA	NA	NA	NA	NA - NA - NA
Ozamiz-Etxebarria et al., 2020 [42] - Spain - 1633	1293 (79.7%) - 426 y	Elementary, Middle and High - NA	808 (49.5%)	NA	526 (32.2%)	NA	NA	NA - NA - NA
Palma-Vasquez, Carrasco, Hernando-Rodriguez, 2021 [43] - Chile - 278	227 (81.65%) - NA - NA	Elementary and Middle - >10 y	NA	NA	162 (58.27%)	NA	NA	NA - NA - NA
Panisoara et al., 2020 [9] - Romania - 988	949 (96.8%) - 20–68 y	NA - NA	NA	NA	NA	NA	NA	NA - NA - NA
Randall et al., 2021 [44] - USA - 1434	1409 (98.3%) - 42 y - White (60%)	Elementary - NA	NA	NA	NA	NA	NA	NA - NA - NA
Seyahi et al., 2020 [45] - Turkey - 851	526 (61.8%) - 35 y - NA	High - NA	232 (29.1%)	363 (42.7%)	373 (42.7%)	NA	140 (15.3%)	NA - NA - NA
Sigursteinsdottir, Rafnsdottir, 2022 [46] - Iceland - 920	759 (82.5%) - 41–50 y - NA	Elementary - 11–20 y	NA	NA	NA	NA	NA	NA - NA - NA
Solís García et al., 2021 [47] - Spain - 640	462 (72,18%) - 43.54 y	Elementary, Middle and High - NA	NA	NA	NA	NA	NA	NA - NA - NA
Souza e Silva et al, 2021 [10] - Brazil - 15,641	12,817 (81.9%) - 41–60 y	Middle and High - 1–10 y	4044 (25.9%)	NA	NA	NA	5047 (32.3%)	NA - NA - NA
Stachteas, Stachteas, 2020 [48] - Greece - 226	143 (63.3%) - <60 y	Middle - NA	NA	NA	100 (44.4%)	NA	NA	NA - NA - NA
Wang et al., 2021 [49] - China - 99,611	41,126 (60.33%) - 18–79 y	Elementary and High - NA	12,100 (32.38%)	NA	NA	NA	NA	NA - NA - NA
Zhou et al., 2021 [50] - China - 11,096	871 (79.5%) - 41 y	High - NA	NA	NA	801 (73%)	NA	NA	57 (5.2%) - NA - NA
Zhou, Yao, 2020 [51] - China - 751	257 (34.2%) - 40.02 y	Elementary and Middle - NA	NA	NA	NA	NA	NA	NA - NA - NA

Legend: NA—Not available; y—Years; M—mean; SD—standard deviation.

## Data Availability

The data presented in this study are available on request from the corresponding author.

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
