# Peer review of "The Impact of the COVID-19 Pandemic on the Mental Health of Teachers and Its Possible Risk Factors: A Systematic Review"

_ijerph, 2023, doi:10.3390/ijerph20031747_

Round 1

Reviewer 1 Report

1. Very relevant manuscript with current theme.

2. I think it's an important matter.

Author Response

Dears Reviewers and Editor,

Thank you very Much for your valuable suggestions.

Please find enclosed our answers.

Yours sincerely,

Authors

REVIEWER 1: approved without corrections.

 -Answer: Thank you very much.

Reviewer 2 Report

The Paper is interesting and well-written. I just have a few but very important suggestions:

Newcastle Ottawa Scale shows lower level of reliability why use it?

Also, I suggest authors somehow address intragroup variability which is very important. There is literature showing that people differently reacted to Covid, for examples see Gamsakhurdia's 2022 paper on covid.

Author Response

Dears Reviewers and Editor,

Thank you very Much for your valuable suggestions.

Please find enclosed our answers.

Yours sincerely,

Authors

REVIEWER 2:

  1. The Newcastle Ottawa scale shows a lower level of reliability. Why use it?
     -Answer: The choice of the NOS Scale is due to the suitability of this scale for cross-sectional studies. As most of our studies are cross-sectional, we chose this one. However, in order to answer your question and also increase the reliability of the analysis, we did a second quality assessment of the papers using the Study Quality Assessment Tool (https://www.nhlbi.nih.gov/health-topics/study-quality- assessment-tools) linked to the National Heart, Lung and Blood Institute (NHLBI);

In addition, I suggest that authors somehow address intragroup variability, which is very important. There is literature showing that people reacted differently to Covid, for example see Gamsakhurdia's 2022 article on covid.
 -Answer: We have added a paragraph in the discussion in which we talk about this inhomogeneity of intragroup response to COVID-19, as well as quoting the suggested author. We also reinforce this aspect when we talk about the limitations of the review.

Reviewer 3 Report

The paper aimed at a very important topic of teachers Mental health during the COVID-19 pandemic. However, the article included in the review are very diverse and I am not sure whether you can draw many overall conclusion from combining them. 

Abstract

-         Unfortunately, I don’t understand the numbers in the following sentence

“High levels of stress (52.38%)/burnout (>50%) were the most prevalent disorders, followed by depression (44.62%)/anxiety (41.45%)”. àwhat does the % tell us and why is there no specific number for burnout?

-          I am unsure how the authors came to the following conclusion from the results presented in the abstract: Therefore, the COVID-19 pandemic was an important trigger of psychiatric pathologies among teachers, with work factors having strong participation in the illness process. There is no comparison for psychological health before and during the pandemic in the papers, so in my opinion this conclusion cannot be drawn from the results presented in the abstract.

1. Introduction

General: please add more substance to the introduction section. Questions that should be answered are: Why should teachers be affected more by the pandemic than other occupations? Why is teachers mental health important? Why is a review in this area important? How was teachers health before the pandemic?

I also would appreciate a change in tone of the introduction section. At the moment the authors imply and state that teachers suffered under the pandemic. But if this was already clear, the presented review would not be necessary.

Please outline the importance and aim of the review in more detail.

Smaller issues:

1.       Line 30: low wages does not apply to all countries

2.       Line 48-50: Please add English publications as well and change the wording for suffering

2. Material and Methods

Please consider that it takes some time between data collection and publication when explaining your timeframe.

I think the beginning paragraphs 3.1-3.3 of the results section would belong into the method section.

3. Results

In general: Authors should be careful in drawing conclusions from only one or two studies. I find some statements too strong for the differences in study design as well as number of participants and region. I would appreciate it, if the number of studies showing a result are included and also studies that did investigate the factor but did not find a significant result are mentioned. Results should be presented more precise. Maybe separate tables with study name, result and influencing factors would help.

Unfortunately, I could not see figure 1. But I assume it was the prisma statement/figure. Which would belong in the Method section.

I don’t really understand why four articles were dismissed due to “four because they presented inconclusive result”. Maybe the authors could elaborate more in this area.

Line 64: What is meant by “resulting labor changes”?

Line 119: Please not that stress is no disorder.

Line 129: please cite authors and use the correct term for the scale. “Maslach Scale” does not exist. However, the Maslach Burnout Inventory does.

3.3. please state which publications scored what point.

Line 144: please explain in the method section how the number was calculated. Same is true for all the other measures of mental health

Line 150: again stress is no disorder

Line 157-158: Authors should be careful in drawing conclusions like this from only two studies.

Line 202: Why isn’t here the same concrete number given as for stress? I also don’t know how the conclusion or percentage was created.

Line 224-226: I don’t understand this sentence

4. Discussion

In general: A discussion of the results is missing. There should be a comparision to other reviews or to reviews that have been done before the pandemic.

Line 311: “The COVID-19 pandemic has increased the prevalence of psychiatric pathologies in teachers.” Statement much too strong. What evidence supports this statement? There were only 4 longitudinal studies that could have indicated this conclusion. In my opinion, the influence of the COVID-19 pandemic on mental health can only be investigated in longitudinal studies.

The limitation section needs to be broadened and explained in more details. Please add also add, differences between countries as limitation, study design.

5. Conclusions

Line 354: please note that neither stress nor burnout is a psychiatric disorder.

Please also add that there is a limited number of studies in this area and there is a need for more to get a full picture and draw definite conclusions.

Author Response

Dears Reviewers and Editor,

Thank you very Much for your valuable suggestions.

Please find enclosed our answers.

Yours sincerely,

Authors

REVIEWER 3: 

1 Abstract

1.1.   Unfortunately, I don’t understand the numbers in the following sentence

“High levels of stress (52.38%)/burnout (>50%) were the most prevalent disorders, followed by depression (44.62%)/anxiety (41.45%)”. àwhat does the % tell us and why is there no specific number for burnout?

-Answer: We appreciate the considerations. As this section was confusing, we chose to remove it.

       1.2.    I am unsure how the authors came to the following conclusion from the results presented in the abstract: Therefore, the COVID-19 pandemic was an important trigger of psychiatric pathologies among teachers, with work factors having strong participation in the illness process. There is no comparison for psychological health before and during the pandemic in the papers, so in my opinion this conclusion cannot be drawn from the results presented in the abstract.

    -Answer: We reformulated the conclusions section in the Abstract.

  1. Introduction

General: please add more substance to the introduction section. Questions that should be answered are: Why should teachers be affected more by the pandemic than other occupations? Why is teachers mental health important? Why is a review in this area important? How was teachers health before the pandemic?

2.1. I also would appreciate a change in tone of the introduction section. At the moment the authors imply and state that teachers suffered under the pandemic. But if this was already clear, the presented review would not be necessary.

Please outline the importance and aim of the review in more detail.

-Answer: We reformulated the introduction section. As well as trying to make the objective clearer and justify the importance of this review.

2.2. Smaller issues:

2.2.1.       Line 30: low wages does not apply to all countries
-Answer: the snippet has been removed.

2.2.2.       Line 48-50: Please add English publications as well and change the wording for suffering.
-Answer: It was made.

  1. Material and Methods

3.1. Please consider that it takes some time between data collection and publication when explaining your timeframe.
-Answer: We redid the quest increasing the collection time to December 1, 2022.

3.2. I think the beginning paragraphs 3.1-3.3 of the results section would belong into the method section.
-Answer: We transferred the sections as instructed above.

  1. Results

In general: Authors should be careful in drawing conclusions from only one or two studies. I find some statements too strong for the differences in study design as well as number of participants and region. I would appreciate it, if the number of studies showing a result are included and also studies that did investigate the factor but did not find a significant result are mentioned. Results should be presented more precise. Maybe separate tables with study name, result and influencing factors would help.

-Answer: Dear reviewer, due to the short period of time (10 days) for reviewing the paper, it has become unfeasible to make this change - the only one of your suggestions not made; because, (1) we had to perform new data mining to update the revision until December 2022; (2) we performed a new analysis of the included studies using a second scale; (3) we placed a new column in Table 1 to highlight the main biases of each study; (4) we had to restructure the other sections. We still started the requested process, but we realized that within each study, there were different methodologies, sets of variables and analysis models. Thus, we would need more time, or even a new study, probably a meta-analysis, to answer such questions. We are deeply sorry and hope to count on your understanding.

3.2. Unfortunately, I could not see figure 1. But I assume it was the prisma statement/figure. Which would belong in the Method section.
-Answer: Figure 1 was uploaded to the system. We don't understand what happened either, however, we put it back.

3.3. I don’t really understand why four articles were dismissed due to “four because they presented inconclusive result”. Maybe the authors could elaborate more in this area.
-Answer: we rephrased the excerpt - these papers had no clear methodology in their scope, so they were excluded.

3.4 Line 64: What is meant by “resulting labor changes”?
-Answer: We removed the excerpt because it was confusing.

3.5 Line 119: Please not that stress is no disorder.
-Answer: We removed the excerpt because it was confusing. Reassessing the papers, we saw that a more appropriate definition would be generalized anxiety disorder.

3.6 Line 129: please cite authors and use the correct term for the scale. “Maslach Scale” does not exist. However, the Maslach Burnout Inventory does.
-Answer: We corrected this term.

3.7. Line 144: please explain in the method section how the number was calculated. Same is true for all the other measures of mental health
-Answer: We removed that part.

3.8. Line 150: again stress is no disorder
-Answer: This part has been corrected.

3.9. Line 157-158: Authors should be careful in drawing conclusions like this from only two studies.
-Answer: We reformulate the conclusion section.

3.10. Line 202: Why isn’t here the same concrete number given as for stress? I also don’t know how the conclusion or percentage was created.
-Answer: We removed this nomenclature and analysis, as it brought confusion to the text.

3.11. Line 224-226: I don’t understand this sentence.
-Answer: The snippet has been removed.

  1. Discussion

In general: A discussion of the results is missing. There should be a comparision to other reviews or to reviews that have been done before the pandemic.
-Answer: We reformulated the discussion section and sought to make comparisons with other reviews and cohorts. Including studies that showed the mental health situation of pre-pandemic teachers.

Line 311: “The COVID-19 pandemic has increased the prevalence of psychiatric pathologies in teachers.” Statement much too strong. What evidence supports this statement? There were only 4 longitudinal studies that could have indicated this conclusion. In my opinion, the influence of the COVID-19 pandemic on mental health can only be investigated in longitudinal studies.
-Answer: we agree. We remove the section.

The limitation section needs to be broadened and explained in more details. Please add also add, differences between countries as limitation, study design.
-Answer: Considering your remarks, we have expanded the final considerations section.

  1. Conclusions

Line 354: please note that neither stress nor burnout is a psychiatric disorder.
-Answer: We believe it is important to maintain the analysis of Burnout, but we separate it from “psychiatric pathologies”, when we refer to it in the text. As well as removing the term 'stress'.

Please also add that there is a limited number of studies in this area and there is a need for more to get a full picture and draw definite conclusions.
-Answer: It is Done.

Round 2

Reviewer 2 Report

No improvement. 

Author Response

Dears Reviewers and Editor,

Thank you very Much for your valuable suggestions.

Please find enclosed our answers.

Yours sincerely,

REVIWER 2 - 2nd round:

1 No improvmente.

Answer: Thank you very much.